# Energy Reserve Allocation in the Trade-Off between Migration and Reproduction in Fall Armyworm

**DOI:** 10.3390/insects15100809

**Published:** 2024-10-16

**Authors:** Chuan-Feng Xu, Peng-Cheng Liu, Jason W. Chapman, Karl R. Wotton, Guo-Jun Qi, Yu-Meng Wang, Gao Hu

**Affiliations:** 1Department of Entomology, Nanjing Agricultural University, Nanjing 210095, China; 2020202039@njau.edu.cn (C.-F.X.); pcliu@njau.edu.cn (P.-C.L.); j.chapman2@exeter.ac.uk (J.W.C.); hugao@njau.edu.cn (G.H.); 2College of Ecology and Environment, YuZhang Normal University, Nanchang 330103, China; 3Centre for Ecology and Conservation, University of Exeter, Penryn, Cornwall TR10 9FE, UK; k.r.wotton@exeter.ac.uk; 4Environment and Sustainability Institute, University of Exeter, Cornwall Campus, Penryn TR10 9FE, UK; 5Guangdong Provincial Key Laboratory of High Technology for Plant Protection, Plant Protection Research Institute, Guangdong Academy of Agricultural Sciences, Guangzhou 510640, China; qigj@gdppri.com; 6Key Laboratory of Surveillance and Management of Invasive Alien Species, Department of Biology and Engineering of Environment, Guiyang University, Guiyang 550005, China; 7Guizhou Education Department, Guiyang University, Guiyang 550005, China

**Keywords:** migration and reproduction, transcriptomics, *Spodoptera frugiperda*, energy reallocation, flight capacity

## Abstract

Insect migratory success is dependent on the correct allocation of energy resources with them opting to allocate energy to fuel flight rather than spend it on reproduction. However, the molecular metabolism underlying the trade-off between migration on one hand and reproduction on the other is poorly understood. Our study provides evidence supporting the trade-off between migration and reproduction in the fall armyworm *Spodoptera frugiperda* (Lepidoptera: Noctuidae) by comparing behavioral and physiological characteristics between migratory and non-migratory individuals. Transcriptome analysis revealed the enrichment of genes with roles in lipid and carbohydrate metabolism in migratory females. Additionally, more triglycerides were stored in the abdomen of migratory females and provided energy for later phases of migratory flight. These findings contribute to a deeper understanding of the physiological strategies employed by migratory insects and set the stage for future research on energy allocation and trade-offs in long-distance migration.

## 1. Introduction

Insect migration is a remarkable phenomenon characterized by seasonal or life stage-related movements often involving extensive long-distance flights [1,2,3]. This behavior serves as a distinctive and adaptive strategy for moving between seasonally varying environments, which is primarily driven by locating new food sources, avoiding adverse conditions, or identifying suitable breeding sites [4,5,6]. Migration requires a substantial energy expenditure to support both flight and essential metabolic processes [7,8,9]. To ensure uninterrupted flight, some migratory insects temporarily inhibit ovarian development due to competition for these limited energy resources [10,11,12]. This trade-off, known as the “oogenesis-flight syndrome” [13,14], leads to the initial migration of female adults predominantly occurring during the “pre-oviposition period” when their ovaries are immature. Conversely, ovarian development is accelerated post-migration [14,15]. This intriguing phenomenon has been observed across various insect species [15,16,17]. The trade-off between migration and reproduction is intimately linked to stored energy.

For insects, the majority of energy substrates used during flight are sourced from the fat body [18]. Among these, glycogen and triglycerides are the primary forms of energy storage in the fat body [19]. When insects initiate flight, glycogen is rapidly converted to trehalose, which is further broken down into glucose, entering pathways like glycolysis to release energy [20]. As long-distance migration progresses, insects switch to using triglycerides to provide energy for longer flights [19], but in the absence of sustained flight, these triglycerides are redirected toward the ovaries for egg development [21]. Consequently, the trade-off between migration and reproduction relies on the allocation of energy and resources. However, our understanding of the molecular mechanisms underlying this trade-off remains limited.

The fall armyworm, *Spodoptera frugiperda* (J.E. Smith), is a well-known agricultural pest native to tropical and subtropical regions of the Americas [22]. This insect exhibits a broad host range and undertakes seasonal migration between the southern and northern regions of North America every year, causing significant economic damage to major food crops [22]. Over the past few years, the fall armyworm has invaded China and seasonally establishes in most areas of eastern China [23,24]. Previous studies showed that a period of flight significantly shortened the pre-oviposition period, suggesting ovarian development can be accelerated by flight in fall armyworms [25]. However, it is not yet clear whether migratory individuals have more immature ovaries before flight than non-migrants or what transcriptomic changes underly this transition.

In this study, we performed a migratory assay to distinguish migratory individuals in the population of fall armyworms and predict that these individuals will display the oogenesis-flight syndrome. To investigate this, we utilize a combination of morphological analysis of ovaries, measurements of fat and carbohydrate levels and tethered flight to characterize the distribution of traits in migratory vs. non-migratory individuals along with transcriptomic analysis to identify the associated differentially expressed genes.

## 2. Materials and Methods

### 2.1. Insect Rearing

The population of fall armyworm pupae was collected from Yuanjiang County, Yunnan, southwest China, in June 2021. Subsequently, they were transported to our laboratory and reared in an artificial climate chamber (Ningbo Jiangnan Instrument Factor) at 25 ± 1 °C under a photoperiod of 14: 10 h for over ten generations. Larvae were provided with artificial feed in individual feeding tubes as described in Wang et al. [26]. Newly emerged adults were transferred to 350 mL transparent plastic cups, where cotton wool soaked in a 10% honey solution was placed at the bottom of each cup.

### 2.2. Observation of Migratory Behavior

Field observations have revealed that almost all migratory individuals of another migratory pest moth (*Cnaphalocrocis medinalis*) took off at a vertical angle in the twilight [27]. We thus performed a takeoff assay to evaluate the migratory propensity of the fall armyworm population we held in culture. Individuals that take off vertically toward the top of the cage are considered migratory (MG), distinguishing them from non-migratory (NMG) individuals. The experiments commenced at dusk. All two-day-old adults were placed in a transparent polyvinyl chloride cage (diameter = 60 cm, height = 120 cm, 5~10 individuals per cage) at least 1 h before the test to adjust to the surrounding environment (25 °C, 70% relative humidity). To simulate sunset conditions indoors, we installed 12 parallel fluorescent lamps with adjustable light intensity, and the light intensity was gradually reduced from 1000 lx to 0.1 lx within 30 min. After the lights were turned off, observations of adult takeoff behavior were conducted for 30 min, and non-migratory and migratory individuals were collected for further analysis.

### 2.3. Tethered Flight Test and Ovarian Dissection

The flight performance of the fall armyworms after takeoff was examined using a tethered flight mill [28]. Firstly, the adults were immobilized briefly by placing them in a freezer at −20 °C for 2 min. We then carefully unfolded the wings of the adult insects, gently brushed the scales at the intersection of the abdomen and thorax, and used Pattex glue (Henkel Group) to secure them to the tether. Once the adults had fully recovered, we attached the undamaged individual to the end of the flight arm and recorded their flight duration for 10 h. Finally, tethered adults were collected for the measurement of the energy substrate. All experiments were conducted in complete darkness, maintaining a constant temperature of 25 °C and a relative humidity of 70%.

To compare the ovarian development status among individuals with different migratory behaviors, we dissected female adults at 0.5 h or at 18 h under a stereo microscope (Motic SMZ-161) after the observation of migratory behavior. We assessed the developmental status of the ovaries as set out in [29] and classified them as level II (almost no mature eggs in the ovariole) or level III (a large number of mature eggs in the ovariole). Females were dissected to determine the developmental status of the ovaries 18 h after the takeoff assay and were divided into four groups: MG-II (migratory individuals with ovaries at level II), MG-III (migratory individuals with ovaries at level III), NMG-II (non-migratory individuals with ovaries at level II) and NMG-III (non-migratory individuals with ovaries at level III) according to the dissected ovarian levels of development. Each group consisted of five adult insects collected as samples for RNA sequencing analysis. All samples were promptly frozen in liquid nitrogen to preserve RNA integrity. Three replicates were conducted for each treatment group (MG-II, MG-III, NMG-II, and NMG-III), resulting in a total of twelve adult samples collected.

### 2.4. RNA Sequencing and Transcriptomic Analysis

Total RNA was isolated from the previously collected samples using the Trizol Reagent (Takara, Dalian, China) following the manufacturer’s instructions. We assessed the quality and purity of the RNA using a Nanodrop spectrophotometer (Themo Fisher, Waltham, MA, USA) and 1% agarose electrophoresis. Subsequently, we synthesized cDNA, prepared libraries, and conducted sequencing on the DNBSEQ platform through Shenzhen BGI Co., Ltd., (Shenzhen, China) with a sequencing length of 150 bp. The raw data were submitted to NCBI (NCBI, Bethesda, Rockvile, MD, USA) with the accession number PRJNA1106078.

After sequencing, we filtered the data to isolate high-quality reads by removing sequences with adapters, low-quality bases, and unknown bases. We carried out a comparative analysis with the *S. frugiperda* reference genome database (https://www.ncbi.nlm.nih.gov/datasets/genome/GCF_023101765.2/, accessed on 26 August 2022). The quality of these reads was assessed by calculating the Q20, Q30, GC content, and sequence repeat levels. Gene expression levels for each sample were quantified using RSEM 1.3.1 [30]. To evaluate the consistency among biological replicates, we calculated Pearson’s correlation coefficient. DEGs were determined for each treatment utilizing the DEseq2 R software 4.3.3 package [31]. Gene ontology enrichment analysis of DEGs was carried out using the phyper function in R software 4.3.3, focusing on the Gene Ontology (GO) and Kyoto Encyclopedia of Genes and Genomes (KEGG) databases [32]. Significance was determined when the FDR-corrected *p*-value was <0.05.

### 2.5. Triglyceride and Glycogen Measurement

To compare the energy storage in *S. frugiperda* females after tethered migratory flight, the abdomen and thorax (without wings and legs) were dissected from individuals and used separately to determine triglyceride and glycogen contents. Individual samples were measured separately for triglyceride and glycogen content post-flight. However, when comparing migratory and non-migratory individuals, tissues from three individuals per condition were pooled for analysis.

Triglyceride contents were measured using the triglyceride assay kit (Nanjing Jiancheng Bioengineering institute, Nanjing, China, A110-1-1). The tissue (abdomen and thorax) was homogenized in 500 μL normal saline at low temperature. The lysate was then centrifuged at 587× *g* for 10 min at 4 °C. A 2.5 μL aliquot of the diluted supernatant was used to determined triglyceride content with 250 μL of detection buffer. Mixed solutions were incubated for 10 min at 37 °C and measured using SpectraMax M5 with a wavelength of 510 nm. Since the measured values of triglyceride content needs to be normalized to the protein level in each homogenate, the protein contents of each sample were measured in parallel using the total protein assay kit (Nanjing Jiancheng Bioengineering institute, Nanjing, China, A045-1-2). A 10 μL aliquot of supernatant was used to determine the protein content with 250 μL of detection buffer. Mixed solutions were incubated for 30 min at 37 °C and measured using a SpectraMax M5 with a wavelength of 562 nm.

Glycogen contents were determined using the glycogen assay kit (Comin, Suzhou, China, TY-1-Y). Samples were homogenized in extracting solution (0.1 g sample/500 μL extracting solution) and placed in a water bath of 95 °C for 20 min, during which it was shaken every 5 min. The lysate was diluted with distilled water to 5 mL when cooled. Mixed solutions were centrifuged at 8000× *g* for 10 min at 25 °C. A 60 μL aliquot of supernatant was mixed with 240 μL of detection buffer and incubated for 10 min in a water bath of 95 °C. Finally, 200 μL of reaction mix was measured using a SpectraMax M5 with a wavelength of 620 nm when it cooled to room temperature.

### 2.6. Statistical Analyses

For migratory behavior and ovarian development, differences between groups were assessed by Chi-square test. A Mann–Whitney U test was used for comparing flight traits between different groups. One-way ANOVA or *t*-tests were used for the analysis of energy substrate levels. IBM SPSS Statistics 23.0 was used for statistical analysis.

## 3. Results

### 3.1. Migratory Females Display Delayed Ovarian Development but Higher Flight Performance

Comparisons of the flight performance between MG and NMG individuals demonstrated significantly higher flight duration and flight speed in the MG female group (Figure 1A). However, there were no significant differences between males (Appendix A). As only females exhibited a higher flight capacity, we focused exclusively on female adults in our subsequent experiments.

To investigate the relationship between migratory behavior and ovarian development, we compared the proportion of individuals with ovaries at level II between the two time points. In NMG individuals, we observed a trend of decreasing numbers of individuals with level II ovaries, although this fell just outside of the significance cut-off (40.6% and 21.4% at 0.5 and 18 h, respectively; df = 1, *p* = 0.055), indicating a progression of ovarian maturation. In contrast, similar percentages of level II ovaries were observed at these two time points in the MG group (45.2% and 39.3%, respectively; df = 1, *p* = 0.486) (Figure 1B), indicating a delay of ovarian maturation in the migratory individuals.

### 3.2. Transcriptome Analysis of Migratory and Non-Migratory Females with Different Levels of Ovarian Development

To explore the regulatory mechanisms of this trade-off, we conducted a transcriptome analysis of migratory and non-migratory females with different levels of ovarian development (Figure 2A). A total of 498,975,782 raw reads were obtained from 12 samples. Each sample group generated over 6 GB of high-quality data, with Q30 values ranging from 91.17% to 92.02%, GC content ranging from 41.62% to 43.08%, and unique mapped ratio from 85.19% to 87.43% (Appendix A). These data confirm the high quality of the sequencing data.

To identify transcripts involved in ovarian maturation, we compared NMG-III vs. NMG-II and MG-III vs. MG-II. Differentially expressed genes (DEGs) were determined for each treatment based upon adjusted *p*-value < 0.05 and the absolute value of log_2_ ratio > 1. We identified 29 and 74 up-regulated DEGs as well as 141 and 52 down-regulated DEGs, respectively (Appendix A). To identify transcripts associated with migration behavior and migration, we compared the matched ovarian levels of MG vs. NMG individuals: MG-II vs. NMG-II and MG-III vs. NMG-III. We identified 71 and 159 up-regulated DEGs along with 110 and 104 down-regulated DEGs, respectively (Appendix A).

To characterize the gene set associated with ovarian development, we performed GO enrichment analysis of the DEGs in NMG-III vs. NMG-II and MG-III vs. MG-II. These DEGs were significantly enriched in GO terms as the fatty acid biosynthetic process, fatty acid elongase activity, alcohol-forming fatty acyl-CoA reductase activity, long-chain fatty-acyl-CoA metabolic process, glucose binding, protein kinase activity, nucleoside metabolic process, and UDP-N-acetylgalactosamine transmembrane transport (Figure 2B). Significantly enriched GO terms associated with the migratory-related genes were found by analyzing DEGs in MG-II vs. NMG-II and MG-III vs. NMG-III, and they included the fatty acid biosynthetic process, fatty acid elongase activity, alcohol-forming fatty acyl-CoA reductase activity, long-chain fatty-acyl-CoA metabolic process, glucose binding, cytoplasm, cytoplasmic vesicle, and aminopeptidase activity (Figure 2B). Interestingly, GO terms both significantly enriched in NMG-III vs. NMG-II and MG-III vs. NMG-III were the fatty acid biosynthetic process, fatty acid elongase activity, alcohol-forming fatty acyl-CoA reductase activity, long-chain fatty-acyl-CoA metabolic process, and glucose binding (Figure 2B).

In the comparison of MG-III vs. NMG-III, the primarily enriched KEGG pathways of DEGs included fatty acid metabolism, phototransduction, and carbohydrate digestion and absorption (Figure 2C). In fatty acid metabolism, only hydroxysteroid 17-beta dehydrogenase 12 (*HSD17B12*) exhibited a down-regulated expression in MG-III, whereas other genes were up-regulated. Within the enriched phototransduction pathway, all genes were down-regulated in MG-III. Carbohydrate digestion and absorption pathways included hexokinase (*HK*), alpha-Amylase (*AMY*), Na^+^/K^+^ transport ATPase (*NKA-α*), and phosphatidylinositol phospholipase C, beta (*PLC-β*). The expression of *PLC-β* and *NKA-α* was significantly down-regulated in MG-III, whereas *HK* and *AMY* were significantly up-regulated. These genes exhibited similar differences in the comparison of MG-II vs. NMG-III as observed in the comparison of MG-III vs. NMG-III. However, in the comparison of MG-II vs. MG-III, there was no significant difference in the expression levels of these genes (Figure 2C).

To further explore genes associated with migratory behavior, we identified functions such as locomotion and insulin signaling that may also play a role in regulating migratory behavior in the fall armyworm females. Specifically, we observed the expressions of four genes that were up-regulated in MG females and those of six genes that were down-regulated (Table 1).

### 3.3. Triglyceride Levels Are Increased in the Abdomen of Migratory Individuals

To clarify the association between migratory behavior and energy substrates in fall armyworms, we measured the concentrations of triglycerides and glycogen in the thorax and abdomen of females. Compared to NMG females, MG females stored more triglycerides in the abdomen but not in the thorax (Figure 3A). In contrast, the concentration of glycogen in the thorax of MG females was significantly lower than that of NMG females, while there was no significant difference observed in the abdomen (Figure 3B). Additionally, there was no significant difference in the thorax and abdominal mass between MG and NMG females (Figure 3C).

Furthermore, we compared the concentrations of triglycerides and glycogen in the thorax and abdomen of females with ovaries at level II and III. Surprisingly, no significant difference in these energy reserves was observed, whether in the thorax or abdomen (Appendix A), despite ovaries at level III having a higher weight (Appendix A).

### 3.4. Triglycerides and Glycogen Are Utilized during Flight

To elucidate the dynamics of energy substances during the flight of the fall armyworms, we conducted a regression analysis of the relationship between energy substrate concentrations and flight traits. These results indicated a moderate decrease in abdominal triglyceride concentration during short-distance or short-duration flights. However, with extended flight distances or durations, triglycerides in the abdomen showed a pronounced reduction. Interestingly, no matter how long flight distances and durations were, the concentration of triglycerides in the thorax remained relatively stable (Figure 3D). In contrast, glycogen in the thorax and abdomen declined sharply after brief flights and then exhibited a gentler pattern of decline with increasing flight distance and duration (Figure 3E).

## 4. Discussion

In many migratory insects, individuals that engage in sustained migratory flights tend to inhibit ovarian development, while those that refrain from prolonged flight tend to exhibit enhanced early reproductive capacity [11,15,17]. In this study, we provide evidence supporting the trade-off between migration and reproduction in fall armyworms with migratory females showing a higher flight performance associated with delayed ovarian development. To investigate the regulatory mechanism of this trade-off, we conducted transcriptome analyses on *S. frugiperda* females with varying ovarian development levels and migratory flight capacity. We found that the DEGs in individuals with distinct migratory behaviors and ovarian development levels were primarily linked to energy metabolism. In the following sections, we discuss the possible roles of these enriched pathways in the trade-off between migration and reproduction.

### 4.1. Down-Regulated Genes Associated with Fatty Acid Synthesis May Be a Key Trait of Migratory Females with Ovaries at Level III

For insects, lipid metabolism is essential for growth and reproduction and provides energy needed during ovarian development [19]. Fatty acids play essential roles in lipid metabolism and are required for synthesizing triglycerides, membrane phospholipids, and bioactive lipids [43]. Fatty acid precursors synthesis is catalyzed by fatty acyl-CoA reductases, and the disruption of this gene reduces lipid metabolism activity in *Adelphocoris suturalis* (Hemiptera: Miridae), leading to a significant inhibition of ovarian development [44]. Therefore, the ovarian development of insects is closely related to fatty acid synthesis. In *Gryllus bimaculatus* (Orthoptera: Gryllidae), the lipogenic activity of the fat body sharply increases after emergence and peaked on day 2, which was followed by a gradual decrease due to the vitellogenic oocyte growth [45]. Similarly, we found that the DEGs between NMGIII and NMGII were significantly enriched in the fatty acid biosynthetic process, fatty acid elongase activity, alcohol-forming fatty acyl-CoA reductase activity, and long-chain fatty-acyl-CoA metabolic process, with all related genes down-regulated in NMGIII. This suggests lower fatty acid synthesis ability in females with ovaries at level III compared to level II. Concurrently, the ovary weight of females with ovaries at level III was higher than those of level II, while there was no difference in triglyceride concentration. Similar patterns were observed in *Spodoptera litura* (Lepidoptera: Noctuidae) [21]. We hypothesize that females cease excessive lipid synthesis after oocyte maturation.

### 4.2. Lipids Are Accumulated More in Migratory Moths and Stored as Fuel for Later Phase of Migratory Flight

In many migratory insects, lipid is a major fuel for the later phase of migratory flight [46,47]. The synthesis of fat in insects mainly converts carbohydrates, fats, and proteins in food into triglycerides, phospholipids, and cholesterol through metabolism. The initial step in de novo lipid synthesis is catalyzed by acetyl-CoA carboxylase (*ACACA*), wherein the product, malonyl-CoA, is utilized by fatty acid synthase (*FAS*) to produce palmitic acid, which is a long-chain fatty acid [48,49]. Subsequently, palmitic acid is converted into palmitoyl-CoA, which is regulated by long-chain fatty acid coenzyme A ligase (*FACL*) and palmitoyl-protein thioesterase (*PPT*) [50,51]. Finally, palmitoyl-CoA is transferred to the endoplasmic reticulum, where subsequent fatty acid elongation is carried out by genes such as stearoyl-CoA desaturase (*SCD*), elongase of very-long-chain fatty acid (*ELOVL4*, *7*), *HSD17B12*, and very-long-chain acyl-CoA dehydrogenase (*VLCAD*) [52,53,54]. The absence of certain genes, notably *ACACA*, *FAS*, *FACL*, and *PPT*, can significantly impact the lipid content in adult insects [50,51,55,56].

In this study, *SCD*, *FAS*, *PPT*, *ELOVL4*, *7*, *VLCAD*, *ACACA*, and *FACL* were significantly up-regulated in MG females compared to NMG females, providing a potential pathway for lipid accumulation in migratory moths. To verify this hypothesis, we measured the concentration of triglycerides in the thorax and abdomen of both NMG and MG females. Our results indicate that MG female moths accumulate more triglycerides in the abdomen, providing a foundational energy reserve for extensive migration. In addition, measurements of triglyceride concentrations in moths with different flight durations and distances showed that triglycerides were utilized for migratory flight in a later phase.

### 4.3. Glycogen Is Utilized during the Initial Phases of Flight in Migratory Females

Glycogen serves as a major energy reserve for locomotor activity in insects, including flight [57]. Insects utilize glycogen as fuel for migratory flights across various stages. During this process, glycogen is hydrolyzed into glucose by *AMY* [58], and glucose is then converted into glucose-6-phosphate by *HK* [59]. The up-regulation of *HK* and *AMY* in MG females suggests rapid glycogen utilization compared to NMG females. This hypothesis is supported by assessing glycogen content in individuals flying at different distances and durations. We found that the energy expended during the initial stages of flight primarily originates from glycogen as amounts declined rapidly in the abdomen.

In addition, we observed an up-regulation of *PLC-β* and *NKA-α* in NMG-III females compared to MG-III. *PLC-β*, a membrane-associated enzyme activated by membrane receptors, is considered a crucial mediator in enhancing insulin secretion [60]. The overexpression of *NKA-α*, a key transmembrane protein crucial for maintaining intracellular K^+^ concentration, effectively inhibits insulin secretion [61]. Consequently, insulin levels in NMG-III individuals might be higher than those in MG-III individuals. By measuring the glycogen content, we observed higher concentrations in the thorax of NMG females compared to MG females. This suggests that NMG females continuously consume honey solution before measurement, leading to insulin secretion in response to high glucose levels.

### 4.4. Other Functions Involved in Regulating Individual Migratory Behavior

Light is crucial for insect migration and navigation [62], with phototransduction converting light into electrical signals in photoreceptor cells [63]. Alterations in this pathway can affect light sensitivity [64]. In this study, the phototransduction pathway was enriched in the comparison of MG-III vs. NMG-III with genes involved in signal transduction and response termination down-regulated in MG females. The down-regulation of these genes partly reduced the photosensitivity of insects [65,66]. Research has shown that the phototaxis may be temporarily inhibited after takeoff in *Nilaparvata lugens* (Hemiptera: Delphacidae) [67,68]. Therefore, this adjustment might confer advantages for migratory behavior of fall armyworms at dusk. In this process, insects regulate behavior through signal transduction pathways [69]. Torso-like protein (*TSL*) and beta-arrestin were up-regulated in MG females, which indicating their importance in migratory behavior by influencing insulin signaling and exploratory activity [33,38,70,71]. During the process of signal transduction, insects utilize neurotransmitter like gamma-aminobutyric acid (*GABA*) to regulate neural activity, adapting to different environmental conditions [72]. The *GABA* type B receptor (*GABAB*) was down-regulated in MG females, which suggested its role in migratory behavior [34].

## 5. Conclusions

For migratory insects, it is necessary to regulate the allocation of energy resources based on available reserves and external environmental conditions during long-distance migrations. Our study reveals that migratory fall armyworms exhibit higher flight ability and delayed ovarian development compared to non-migratory females, suggesting that fall armyworms may allocate more energy resources toward flight activity by inhibiting ovarian development during migration. Additionally, the up-regulated genes associated with fatty acid synthesis and glycogen synthesis support a higher energy demand for flight seen in migratory individuals, ensuring sufficient amounts of both lipid and glycogen are available. Consistent with this result, we find more triglycerides stored in the abdomen of migratory females, providing energy for later phases of migratory flight. Our study reveals a trade-off in allocating abdominal triglycerides between migration and reproduction during fall armyworm flight, which may be regulated by energy metabolism related genes. Future investigations should focus on uncovering the genetic mechanisms involved using functional genomic approaches and characterize how this trade-off is achieved at the molecular level in other insect models of migration.

## Figures and Tables

**Figure 1 insects-15-00809-f001:**
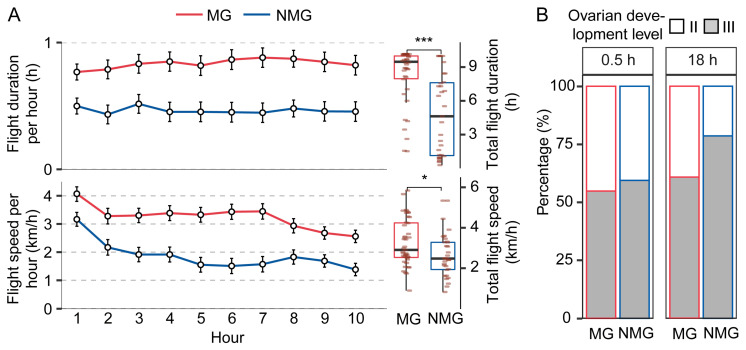
The ovarian development and flight trait divergence between migratory and non-migratory females in *S. frugiperda*. (**A**) The hourly flight duration, flight speed, total flight duration, and flight average speed in females within 10 h tethered flight. MG and NMG represent migratory and non-migratory individuals, respectively. Solid black bars represent medians, boxes represent the interquartile range, whiskers extend to observations within ±1.5 times the interquartile range, * *p* < 0.05, *** *p* < 0.001. Each dot represents a single individual (*n* = 78, Mann–Whitney *U* test). (**B**) The comparison of diverse ovarian percentages in females at different times. 0.5 h and 18 h means the time of dissection after the migratory experiment.

**Figure 2 insects-15-00809-f002:**
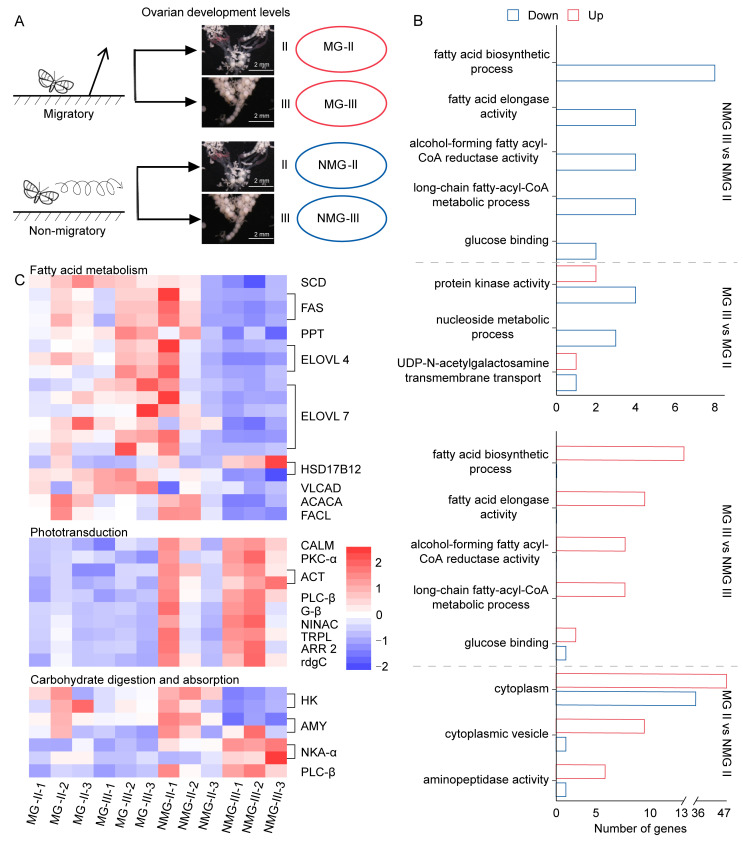
Transcriptome analysis of migratory and non-migratory females with different levels of ovarian development in *S. frugiperda*. (**A**) Schematic diagram for experimental sample collection. Females are divided into migratory (red) and non-migratory (blue) groups based on the takeoff angle after simulating indoor sunset. Then, 18 h later, they were dissected and divided into four groups: MG-II, MG-III, NMG-II and NMG-III according to the ovarian levels. (**B**) The significant enrichment GO terms for DEGs were between NMG-II and NMG-III, MG-II and MG-III, NMG-III and MG-III, and NMG-II and MG-II. (**C**) Heat map based on the expression levels of DEGs in the enriched KEGG pathway between NMG-III and MG-III. These KEGG pathways included fatty acid metabolism, phototransduction, and carbohydrate digestion and absorption based on pathway maps in the KEGG pathway database. Expression values of each gene were normalized (*Z*-score) prior to the analysis. The color blocks indicate the deviation from the gene’s average across all samples.

**Figure 3 insects-15-00809-f003:**
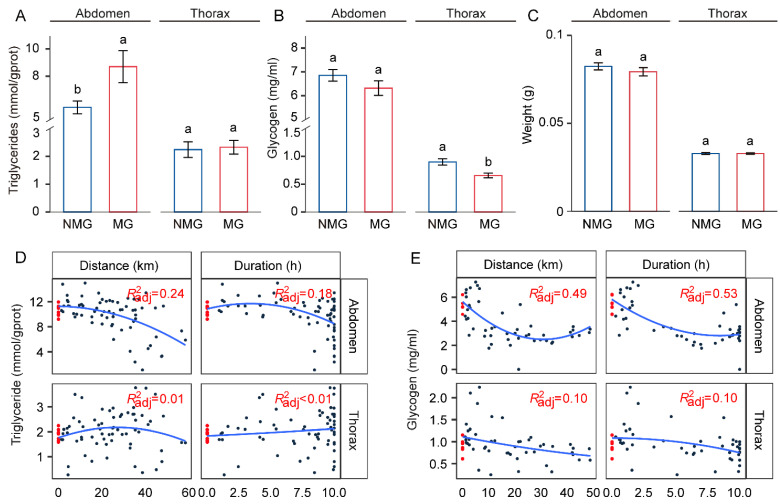
Analysis of energy substance content in *S. frugiperda* females. (**A**–**C**) Comparison of triglycerides (**A**) and glycogen (**B**) levels between migratory and non-migratory individuals in *S. frugiperda* females. (**C**) The weight of female adults was measured as parallel. MG and NMG represent migratory and non-migratory individuals, respectively. Data were presented as mean values ± SEM; the different lowercase letters above bars indicate significant difference (*n* ≥ 6, Student’s *t* test). (**D**,**E**) Dynamics of energy reserve contents in females after tethered flight. Fitting the trends of triglycerides (**D**) and glycogen (**E**) in the thorax and abdomen after 10 h of tethered flight treatment. Solid blue lines indicate the fitted curves (quadratic polynomial) of triglycerides and glycogen in each segment with red labels denoting the degree of fitting for trend lines. Each dot represents an individual with red dots indicating individuals that were not subjected to forced flight (**D**): *n* = 74; (**E**): *n* = 55.

**Table 1 insects-15-00809-t001:** Identification of DEGs possibly associated with migratory behavior of females in *S. frugiperda*.

Gene ID	Annotation	log_2_ FC		Function
		MG-II vs. NMG-II	MG-III vs. NMG-III	
LOC118262805	beta-arrestin	1.25	/	Locomotion [33]
XLOC_008866	gamma-aminobutyric acid type B receptor	−1.77	/	Locomotion [34]
LOC118263634	gamma-aminobutyric acid receptor subunit beta	/	−2.02	Learning and memory [35]
LOC118265756	mevalonate kinase	1.07	/	Juvenile hormone synthesis [36]
LOC118270612	actin-binding LIM protein	−1.22	/	Muscle contraction [37]
LOC118271205	torso-like protein	/	2.01	Insulin signaling [38]
LOC118272993	NADH dehydrogenase [ubiquinone] 1 alpha subcomplex assembly factor 1	/	1.11	Mitochondrion metabolism [39]
LOC126913043	cytochrome c oxidase subunit 1	/	−29.06	Mitochondrion metabolism [40]
LOC118278287	glutamate receptor ionotropic, NMDA 1	/	−2.08	Reproduction [41]
XLOC_002619	prostaglandin-H2 D-isomerase/glutathione transferase	−1.54	−1.71	Reproduction [42]

FC means fold change. “/” indicates no significant differences. Adjusted *p*-value < 0.05 and the absolute value of log_2_ ratio > 1 were used as the threshold to judge the significance of gene expression difference. MG-II and MG-III means migratory females with ovarian development levels of II and III, respectively; NMG-II and NMG-III means non-migratory females with ovarian development levels of II and III.

## Data Availability

All data generated or analyzed during this study are included in this published article [and its Appendix A]. The transcriptomic data supporting the results of this study have been deposited in the NCBI Short Read Archive (SRA) database (PRJNA1106078).

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
