# Peer review of "Energy Reserve Allocation in the Trade-Off between Migration and Reproduction in Fall Armyworm"

_insects, 2024, doi:10.3390/insects15100809_

Round 1
Reviewer 1 Report
Comments and Suggestions for Authors
This is an interesting manuscript on resource allocation in a migratory moth. The authors compared individuals classified as migrants to non-migrants, subjected them to experimental flight, and assessed energy reserves in the body and performed a gene expression study. The results showed that individuals classified as migrants were more active fliers and had less developed ovaries compared to non-migrants, pointing to the presence a trade-off between flight and reproduction. Migrants had a higher triglyceride content in the abdomen than non-migrants, highlighting the role of fats as a fuel for migratory insects. Indeed, abdominal triglyceride levels decreased steadily as flight duration increased, whereas glycogen reserves were quickly depleted. The transcriptome analysis revealed differences among migrants and non-migrants as well as different reproductive stages. These results help to identify genes that may play a role in regulating the trade-off between investment in reproduction vs. investment in energy storage and flight capacity.
I found the study well designed, and the presentation of the methods and results was to a large degree clear. I do have some questions and comments, as there were some sections where I think more information is needed or the presentation could be made clearer.
Line 86: What life stage were the collected insects at? Larvae? Did you rear them for multiple generations before the experimental individuals emerged? Please provide more details.
Line 121: I don’t think it’s correct to call the experimental groups ‘treatments’ if they were assigned to groups according to the status of their ovaries. Calling them ‘groups’ should be fine. Nevertheless, you should describe here what the groups mean (MG-II vs MG-III etc.).
Line 152: Do you mean tissues from three individuals were pooled for the analysis?
Figure 1: The x axis label ‘Flight times (h)’ isn’t clear. Maybe simply ‘Hour’ would work or perhaps ‘Time since initiation of flight (h)’ or something along those lines.
Lines 197-199: More of minor point, the result is said to be significant while the P value is 0.055, i.e. higher than 0.05. What was the significance level?
Figure 3 legend, line 289: “Each dot represents an individual…” I’m a bit confused, based on the Methods section, I understood samples from three individuals per tissue were pooled at least in the analysis of glycogen content. Please clarify.
Line 334: What does ‘ovarian quality’ mean?
Minor points
Lines 57-58: I think it’s unnecessary to point out that only a small amount of the energy expended during flight comes from the flight muscles. Apart from short bursts of activity, in pretty much all animals the majority of energy used is stored outside the muscles. Maybe just point out that the majority of energy substrates are stored in the fat body.
Line 191: “that the” -> “at the”
Lines 325 & 327: I think it would be helpful for the reader to provide the taxonomic groups of the insect species mentioned in the Discussion.
Lines 398, 399 & 407: The abbreviation ‘FAW’ is used here but not elsewhere in the manuscript. Perhaps use the full species name here if there’s no need to abbreviate it.
Comments on the Quality of English LanguageThe Figure 2 legend has a couple of unclear sentences:
"Schematic diagram for selecting experimental samples collection."
I'm not sure what this means: "Schematic diagram for selecting experimental samples" or "Schematic diagram for experimental sample collection."?
Also the next sentence sounds awkward: "Females be divided into group of migratory..."
Author Response
Comments 1: This is an interesting manuscript on resource allocation in a migratory moth. The authors compared individuals classified as migrants to non-migrants, subjected them to experimental flight, and assessed energy reserves in the body and performed a gene expression study. The results showed that individuals classified as migrants were more active fliers and had less developed ovaries compared to non-migrants, pointing to the presence a trade-off between flight and reproduction. Migrants had a higher triglyceride content in the abdomen than non-migrants, highlighting the role of fats as a fuel for migratory insects. Indeed, abdominal triglyceride levels decreased steadily as flight duration increased, whereas glycogen reserves were quickly depleted. The transcriptome analysis revealed differences among migrants and non-migrants as well as different reproductive stages. These results help to identify genes that may play a role in regulating the trade-off between investment in reproduction vs. investment in energy storage and flight capacity.
I found the study well designed, and the presentation of the methods and results was to a large degree clear. I do have some questions and comments, as there were some sections where I think more information is needed or the presentation could be made clearer.
Response 1: We appreciate your positive comments.
Comments 2: Line 86: What life stage were the collected insects at? Larvae? Did you rear them for multiple generations before the experimental individuals emerged? Please provide more details.
Response 2: Thanks for pointing out this issue. We collected fall armyworm pupae from Yuanjiang County, Yunnan to establish the population. Before the experiments in this study were carried out, we had reared them for over ten generations. We have added this information in the revised manuscript. Please see lines 87, and 90.
Comments 3: Line 121: I don’t think it’s correct to call the experimental groups ‘treatments’ if they were assigned to groups according to the status of their ovaries. Calling them ‘groups’ should be fine. Nevertheless, you should describe here what the groups mean (MG-II vs MG-III etc.).
Response 3: Thanks for your valuable suggestions. In the revised manuscript, we have replaced “treatments” with “groups” and added descriptions for the meanings of MG-II, MG-III, NMG-II, and NMG-III. Please see lines 124-128.
Comments 4: Line 152: Do you mean tissues from three individuals were pooled for the analysis?
Comments 5: Figure 3 legend, line 289: “Each dot represents an individual…” I’m a bit confused, based on the Methods section, I understood samples from three individuals per tissue were pooled at least in the analysis of glycogen content. Please clarify.
Response 4: Thank you for your feedback. I apologize for any confusion regarding the measurement of energy reserves in S. frugiperda. In our study, individual samples from each female S. frugiperda were measured for triglyceride and glycogen content after tethered migratory flight. However, when comparing storage between migratory and non-migratory individuals, tissues from three individuals per condition were pooled for analysis. In the revised manuscript, we have added the information in the Materials and methods section to avoid any confusion. Please see lines 156-173.
Comments 6: Figure 1: The x axis label ‘Flight times (h)’ isn’t clear. Maybe simply ‘Hour’ would work or perhaps ‘Time since initiation of flight (h)’ or something along those lines.
Response 6: We appreciate your suggestions. We have amended the x axis label accordingly in the revised manuscript. Please see line 194.
Comments 7: Lines 197-199: More of minor point, the result is said to be significant while the P value is 0.055, i.e. higher than 0.05. What was the significance level?
Response 7: Thanks for raising this issue. We have revised the state that the decrease is " a trend of decreasing ". Please see lines 204-209.
Comments 8: Line 334: What does ‘ovarian quality’ mean?
Response 8: Thanks for pointing out this issue. We have revised to “ovary weight”. Please see line 347.
Comments 9: Lines 57-58: I think it’s unnecessary to point out that only a small amount of the energy expended during flight comes from the flight muscles. Apart from short bursts of activity, in pretty much all animals the majority of energy used is stored outside the muscles. Maybe just point out that the majority of energy substrates are stored in the fat body.
Response 9: Thank you for your suggestions. We have revised the description accordingly. Please see lines 58-60.
Comments 10: Line 191: “that the” -> “at the”
Response 10: Done. Please see line 201.
Comments 11: Lines 325 & 327: I think it would be helpful for the reader to provide the taxonomic groups of the insect species mentioned in the Discussion.
Response 11: Thank you for your valuable suggestions. We have added the taxonomic groups of Adelphocoris suturalis in the revised manuscript. Please see line 338.
Comments 12: Lines 398, 399 & 407: The abbreviation ‘FAW’ is used here but not elsewhere in the manuscript. Perhaps use the full species name here if there’s no need to abbreviate it.
Response 12: Thank you for your helpful comments. We have deleted the abbreviation “FAW” and replaced it with "fall armyworm". Please see lines 411, 413, and 420.
Comments 13: Comments on the Quality of English Language
The Figure 2 legend has a couple of unclear sentences:
"Schematic diagram for selecting experimental samples collection."
I'm not sure what this means: "Schematic diagram for selecting experimental samples" or "Schematic diagram for experimental sample collection."?
Also the next sentence sounds awkward: "Females be divided into group of migratory..."
Response 13: Thank you for your feedback. We have revised these sentences to "Schematic diagram for experimental sample collection" and "Females are divided into migratory (red) and non-migratory (blue) groups based on the takeoff angle after simulating indoor sunset.", respectively. Please see lines 233-236.
Reviewer 2 Report
Comments and Suggestions for Authors
This manuscript describes experimental results comparing behavior-based metabolic processes involved in Spodoptera frugiperda migratory syndromes. It is an excellent addition to the literature and the results provide very satisfying details underpinning observed behavioral differences between migratory and non-migratory individuals. While I broadly expected the differences in lipids and other fuels, the evidence supporting reduced sensitivity to light cues was really fun to see. The manuscript is very well written overall. I suggest some minor changes to improve readability.
Some material from methods was repeated in results unnecessarily (lines 178, 193-196, 205-207, etc)
It would be helpful to re-number the figures to avoid confusion, or consider just omitting the first reference to Fig 2.
I think the differences between MG-II and MG-III (e.g. Fig 2a) indicates the number of hours elapsed before dissection, as a way of measuring delay in ovary development. If so, consider changing the names of the groups to be more descriptive, perhaps using elapsed time? If not, then please elaborate more on this.
Author Response
Comments and Suggestions for Authors
Comments 1: This manuscript describes experimental results comparing behavior-based metabolic processes involved in Spodoptera frugiperda migratory syndromes. It is an excellent addition to the literature and the results provide very satisfying details underpinning observed behavioral differences between migratory and non-migratory individuals. While I broadly expected the differences in lipids and other fuels, the evidence supporting reduced sensitivity to light cues was really fun to see. The manuscript is very well written overall. I suggest some minor changes to improve readability.
Response 1: We are very grateful for your kind appraisal. Thank you for your time and dedication in reviewing our manuscript.
Comments 2: Some material from methods was repeated in results unnecessarily (lines 178, 193-196, 205-207, etc)
Response 2: Thank you for your valuable input. In the revised manuscript, we have carefully revised the text and removed these description. Please see lines 187-188, 204-209, 218, and 309-312.
Comments 3: It would be helpful to re-number the figures to avoid confusion, or consider just omitting the first reference to Fig 2.
Response 3: Thank you for your helpful suggestions. We agree with your and have removed the first reference to Fig 2 to avoid any confusion. Please see line 129.
Comments 4: I think the differences between MG-II and MG-III (e.g. Fig 2a) indicates the number of hours elapsed before dissection, as a way of measuring delay in ovary development. If so, consider changing the names of the groups to be more descriptive, perhaps using elapsed time? If not, then please elaborate more on this.
Response 4: Thank you for your insightful comments. We would like to clarify that all dissections were conducted 18 hours after the observation of migratory behavior, and there is no elapsed time difference between the two groups. We have explained this in the legend of Fig 2A. Please see lines 233-236.
Round 2
Reviewer 1 Report
Comments and Suggestions for Authors
Great job with the revision. I only have a few additional comments.
Line 2: Now that I read the title with fresh eyes, I think “Energy reserve allocation…” would be grammatically more correct than “Energy reserves allocation…).
Line 276: Triglycerides levels -> Triglyceride levels
Line 302: analyses -> analysis
Line 327: triacylglycerol -> “triglyceride” is used elsewhere in the manuscript, so probably should be used here as well.
Line 332: Please add the taxonomic details to this species as well.
Line 341: Please add the taxonomic details to this species as well.
Line 391: Please add the taxonomic details to this species as well.
Comments on the Quality of English LanguageSee above
Author Response
Comments and Suggestions for Authors
Comments 1: Great job with the revision. I only have a few additional comments.
Response 1: We appreciate your positive comments.
Comments 2: Line 2: Now that I read the title with fresh eyes, I think “Energy reserve allocation…” would be grammatically more correct than “Energy reserves allocation…).
Response 2: Thanks for pointing out this issue. We have replaced “reserves” with “reserve” to make it grammatically correct. Please see line 2.
Comments 3: Line 276: Triglycerides levels -> Triglyceride levels
Response 3: Done. Please see line 276.
Comments 4: Line 302: analyses -> analysis
Response 4: Done. Please see line 302.
Comments 5: Line 327: triacylglycerol -> “triglyceride” is used elsewhere in the manuscript, so probably should be used here as well.
Response 5: We appreciate your suggestions. We have replaced "triacylglycerol" with "triglyceride" to ensure consistency throughout the manuscript. Please see line 327.
Comments 6: Line 332: Please add the taxonomic details to this species as well.
Line 341: Please add the taxonomic details to this species as well.
Line 391: Please add the taxonomic details to this species as well.
Response 6: Thank you for your valuable suggestions. We have added the taxonomic groups of Gryllus bimaculatus, Spodoptera litura and Nilaparvata lugens in the revised manuscript. Please see lines 332, 341, and 391-392.